# Using Invitation Letters to Increase HPV Vaccination Among Adult Women

**DOI:** 10.3390/curroncol32040216

**Published:** 2025-04-09

**Authors:** Kelly Bunzeluk, Laura Coulter, Donna Turner, Chaeyoon Jeong, Carla Krueger, Austin Hill

**Affiliations:** 1CancerCare Manitoba, Winnipeg, MB R3E 0V9, Canada; lcoulter@cancercare.mb.ca (L.C.); dturner9@cancercare.mb.ca (D.T.); cjeong@cancercare.mb.ca (C.J.); ckrueger2@cancercare.mb.ca (C.K.); ahill6@cancercare.mb.ca (A.H.); 2Department of Community Health Sciences, Faculty of Medicine, University of Manitoba, Winnipeg, MB R3E 0W2, Canada

**Keywords:** human papillomavirus, HPV vaccine, vaccine uptake, elimination of cervical cancer, vaccine interventions

## Abstract

(1) Background: In Manitoba, most people get the HPV vaccine through a publicly funded school-based program. If they miss the school-based program, they remain eligible for the free HPV immunization program. This study explored whether invitation and reminder letters would increase HPV vaccine uptake among adult women who remained eligible for the publicly funded program. (2) Methods: Eligible individuals were randomized into three groups of 4650 women each. Intervention groups I and II were mailed an information package inviting them to be vaccinated. Six weeks later, intervention group II received a reminder letter if they remained unvaccinated. The control group received no correspondence related to HPV vaccination. Vaccination status, defined as at least one dose of an approved HPV vaccine, was calculated six months after the packages were mailed. (3) Results: Overall, 4.0% (3.4–4.6%, *p* < 0.0001) of individuals in intervention group II (invitation/reminder) and 2.5% (2.1–3.0%, *p* < 0.0001) of individuals in intervention group I (invitation) received one dose of the HPV vaccine. Compared to the control group, sending invitations/reminders and invitation packages increased the likelihood of getting at least one dose of the HPV vaccine by 4.9 times (3.4–6.9, *p* < 0.0001) and 3.0 times (2.1–4.4, *p* < 0.0001), respectively. (4) Conclusion: Sending invitation and reminder letters to unvaccinated women may be an effective and low-cost way to increase HPV vaccination coverage among adults who are eligible for the publicly funded immunization program.

## 1. Introduction

The HPV (human papillomavirus) vaccine is one of the most effective ways to prevent cancer. Not only can the vaccine prevent over 90% of cervical cancers, it can also reduce rates of vulvar, vaginal, anal, oropharyngeal, and penile cancers [1], as well as non-cancerous genital warts. In 2015, it was estimated that 3829 cases of cancer in Canada were caused by HPV [2].

Manitoba is a province in central Canada with a population of about 1.34 million people. All registered permanent residents of Manitoba are eligible for health benefits under The Health Services Insurance Act. However, the low population density of 2.5 people per km^2^ [3] makes it challenging for everyone in the population to access health services. In 2008, Manitoba introduced a publicly funded HPV vaccine program for girls born in 1997 or later. Immunizations are primarily offered in grade six (approximate age: 10–11 years) through a school-based program, though families can also access vaccines from their healthcare providers. In 2016, the HPV vaccine program became available to boys born in 2002 or later, and it is also primarily delivered in grade six through a school-based program [4].

Females born in 1997 and later are eligible to receive the HPV vaccine free of charge. Most girls receive the vaccine through the school-based program. However, since Manitoba maintains a once eligible, always eligible policy, individuals can be vaccinated even if they missed the school-based program or their parents opted out of the program on their behalf [5]. This policy is uncommon in Canada; most jurisdictions have an age cut-off for HPV vaccine eligibility [6]. The Manitoba policy, therefore, provides an opportunity to reach eligible individuals once they become adults.

In 2022, Manitoba’s HPV vaccination coverage rate among females at age 17 was 70% [7]. This rate remains below the national target of 90% [8] and varies by region. Among vaccine-eligible adult women (born 1997–2002), 29% have had zero doses of the HPV vaccine [9]. The initial vaccination decisions were made by parents. Now that some of the vaccine-eligible cohort have reached adulthood, they could be informed of their eligibility and access options so that they can make their own informed decision about vaccination.

Invitation letters have been demonstrated to be effective in other health services, such as cancer screening [10,11,12,13,14,15] and immunization [16,17] uptake. However, to our knowledge, invitation letters have not been used to remind adults of their continued eligibility for childhood vaccines. This project investigated the association between mailed invitation and reminder letters with subsequent HPV vaccination among eligible adult women who were unvaccinated.

## 2. Materials and Methods

This study targeted adult women who had received zero doses of the HPV vaccine but were still eligible for the publicly funded HPV immunization program. This included women aged 20 years and older who were born in 1997 or later.

### 2.1. Study Group Selection

The potential study sample was generated from data contained within the CervixCheck registry. CervixCheck is Manitoba’s population-based cervical screening program. It supports organized cervical cancer screening with Pap testing for women age 21–69. CervixCheck’s database registry includes the names and addresses of all adult females in the province, as well as their health histories related to cervical cancer screening and follow-up, HPV vaccination, hysterectomy, and cervical cancer.

A sample size calculation was completed to determine group sizes. To obtain 80% power, a sample size of 1200 individuals, evenly distributed across the three groups, would be needed to detect a difference at a 5% level of significance. However, bigger group sizes were selected to increase the potential HPV vaccine uptake rate.

Approximately 14,000 women who met the study inclusion criteria were randomized into three study groups. Intervention group I included 4650 women who were mailed an invitation letter. Intervention group II included 4650 women who were mailed an invitation letter and, if needed, a reminder letter. The control group included 4650 women who received no correspondence related to HPV vaccination. The protocol is shown in Figure 1.

People were included in the potential study sample if they met the following inclusion criteria:Have had zero doses of an approved HPV vaccine;At least 20 years old and born in 1997 or later;Female sex, as registered with Manitoba Health;Valid Manitoba health insurance coverage and a complete Manitoba mailing address.

People were excluded from the potential study sample if they met the following exclusion criteria:Have received one or more doses of an approved HPV vaccine;Have had cervical cancer;Have had a hysterectomy;In active colposcopy treatment.

The sample groups were derived using random allocation. First, the population of women in the registry who met the study inclusion and exclusion criteria were identified. Then, women were assigned to a study arm using restricted randomization to achieve the desired sample sizes. Random selection was carried out using SAS17 version 9.4 (SAS Institute Inc., Cary, NC, USA).

### 2.2. Mail-Outs

The packages, including their content, design, and branding, were developed before the project started in consultation with public advisors. The invitation package sent to everyone in the intervention groups included a bilingual (English and French) invitation letter and an educational insert in the form of ‘frequently asked questions’. The reminder package included a bilingual reminder letter.

In May 2023, invitation packages were mailed to women in Intervention Groups I and II. In June 2023, reminder packages were sent to women in Intervention Group II who had not yet been vaccinated.

The initial groups included 4650 women each (13,950 total). However, subsequent analyses revealed that 49–51 people in each group (1.1% overall) had incomplete immunization records within the registry (i.e., an HPV vaccine was administered shortly before the groups were created) and an additional 3.2% became ineligible due to mobility or death within the 6-month study period. These individuals were excluded from the final analysis. About 500 (5.5%) people in the intervention groups were also excluded due to returned mail. Returned mail is typical for screening program communications and occurs when people change addresses but do not notify the health insurance registry. For example, 7.8% of cervical screening invitation letters sent to people aged 24–26 were returned in 2023. Final group sizes and reasons for exclusion are shown in Figure 2.

### 2.3. Assessing Intervention Response

To determine whether the invitation letters increased vaccine uptake among adult women, vaccination status was measured six months after the information packages were mailed. The six-month timeframe was selected to provide a time within which the vaccination behavior could potentially be attributed to the invitation letter. This timeframe was based on the relative ease of booking immunizations or attending walk-up appointments within six months, as well as the typical local pattern of scheduling vaccine appointments in primary care clinics within 3–6 months. An additional uptake rate was calculated at twelve months to test whether reduced access to primary care during the study timeframe affected the results.

Women who received at least one dose of an approved HPV vaccine within six or twelve months of the invitation were counted as vaccinated for the analysis. To assess whether reduced access to primary care affected vaccine uptake, a subsequent analysis was carried out, extending the follow-up timeframe to twelve months.

A chi-square test of homogeneity was performed to determine whether HPV vaccination uptake differed across the three study groups, and logistic regression was used to assess the strength of the relationship between the dependent variable (vaccine uptake) and the independent variable (study arm). Analyses were conducted using SAS17 version 9.4 (SAS Institute Inc., Cary, NC, USA).

This study was approved by the University of Manitoba’s Health Research Ethics Board (HS25709, H2022:325) and CancerCare Manitoba’s Research Review Impact Committee (2022-028).

## 3. Results

### 3.1. Cohort Demographics

The group demographics are shown in Table 1. There were no differences between the groups with respect to age, geography, or cervical cancer screening status.

### 3.2. Six-Month HPV Vaccine Uptake

Table 2 shows the results of vaccine uptake within six months of the invitation mailing date. Of the individuals who received an invitation and reminder letter, 4.0% received one dose of the HPV vaccine. Of the individuals who only received the invitation letter, 2.5% received one dose of the HPV vaccine, compared to only 0.9% of the control group. The difference between the groups was statistically significant at a 5% level of significance (X^2^ = 91.1, *p* < 0.0001).

Logistic regression was completed to determine the effect size (see Table 3). Compared to the control group, sending an invitation package increased the likelihood of getting one dose of the HPV vaccine by 3.0 times. Sending invitation and reminder packages increased the likelihood of getting the vaccine by 4.9 times compared to the control group.

### 3.3. Twelve-Month HPV Vaccine Uptake

The vaccine uptake pattern observed over six months was the same as what was observed over twelve months: individuals who received the invitation and reminder letters were most likely to get an HPV vaccine, followed by individuals who only received an invitation letter, and then those in the control group. All groups had higher proportions of HPV uptake over 12 months compared to 6 months (see Table 4). The differences between the intervention and control groups remained statistically significant at a 5% level of significance (X^2^ = 78.9, *p* < 0.0001).

Logistic regression was completed to determine the effect size within 12 months (see Table 5). Compared to the control group, sending an invitation package increased the likelihood of getting one dose of the HPV vaccine by 2.3 times. Sending invitation and reminder packages increased the likelihood of getting the vaccine by 3.2 times compared to the control group.

### 3.4. HPV Vaccine Uptake by Geographic Region

The HPV vaccine uptake pattern observed over six months was similar to that observed over 12 months (Table 6), with the difference between urban and rural areas decreasing between 6 and 12 months post invite. This may indicate that people in rural areas required more time for HPV vaccination compared to those in urban areas.

## 4. Discussion

The World Health Organization has called for the elimination of cervical cancer through improvements in three key areas: HPV vaccination, cervical cancer screening, and treatment [18]. This project explored whether invitation and reminder letters to unvaccinated adult women could impact the first area by increasing HPV vaccine uptake.

Unvaccinated individuals, namely those who were not vaccinated in the publicly funded school-based program, were targeted for the study. The reason for their vaccination status is not known; it could have been the result of an intentional decision by their parents/guardians (e.g., based on their values or preferences about immunization) or an unintentional decision (e.g., the result of a practical or structural barrier). Additionally, they could have moved into the province at a later age, thereby missing the school-based program.

Regardless of their initial reason for not being vaccinated, this project sought to determine whether reminding unvaccinated adult women of their ongoing eligibility for the publicly funded program could increase HPV vaccination coverage. While Manitoba maintains a relatively unique once eligible, always eligible policy for vaccination, these findings could apply to any jurisdiction that extends eligibility for free HPV vaccines into adulthood (ages 20–25). Further study is needed to determine whether this approach could be integrated into broader immunization programs.

### 4.1. Main Findings

While this study found a relatively low HPV vaccine uptake among the groups (4.0% among people who received invitation and reminder letters and 2.5% among people who only received the invitation letters), a significant difference was observed between the groups. Compared to the control group, who were not sent invitation letters, the invitation/reminder group was 4.9 times as likely to receive one dose of the HPV vaccine within six months, and the invitation group was 3.0 times as likely. This demonstrates that a relatively simple and low-cost intervention can be effective at increasing HPV vaccine rates among unvaccinated adult women.

Since the cervical cancer screening registry contains HPV vaccination information on females 18 years of age and older, this approach could be used routinely to remind unvaccinated individuals of their HPV vaccine eligibility. Given the small increase in vaccine uptake, this approach may not be effective on its own but could be considered alongside other public health promotion activities (e.g., public awareness campaigns). To our knowledge, no immunization campaigns were run during this study timeframe, but routine educational events were held.

The cost of sending letters is very small compared to the potential long-term impact of a well-vaccinated population. The HPV vaccine is highly effective at preventing HPV-related cancers. With good population coverage, the need for follow-up and treatment for cancer and pre-cancerous lesions could be significantly reduced, eventually decreasing morbidity and mortality from HPV cancers.

### 4.2. Strengths and Limitations

One of the key strengths of this study is the robust cervical cancer screening registry in Manitoba. Having access to the population-based registry allowed us to determine vaccine status and eligibility for the publicly funded program and allowed targeted invitation and reminder letters to be sent. Unfortunately, the registry only contains information about adult females. Therefore, even though the HPV vaccine is more effective if given before sexual contact, it was not possible to include children or contact their parents/guardians. Further research is needed to determine whether an invitation strategy would work with younger cohorts.

Even though registry data are updated monthly, it was necessary to adjust for incomplete immunization histories. About 50 records in each cohort were updated after the study groups were identified. This was accounted for by removing the individuals who became ineligible from the analysis. Additionally, over 500 invitation packages were returned due to incorrect addresses. This return rate is typical with screening program communications but might affect the impact of these results in this population. While residents are required to update their addresses with the provincial health registry, increased address mobility and improved health status (less contact with the health system) among this age cohort might explain the returned mail.

Additionally, while the odds ratios were quite large (people in the intervention groups were 4.9 and 3.0 times as likely to get an HPV vaccine compared to the control group), the net uptake in each group was small (4.0% and 2.5% increases). The large study sample size may have exaggerated these small differences and found them to be statistically significant. Regardless, the small increases observed in this study were expected, given that the cohort (or their parents/guardians) had likely rejected a vaccine offer on at least one previous occasion.

Finally, there was no controlling for potential confounding variables in this study. For example, factors such as proximity to healthcare services, rurality, and past healthcare use may have influenced someone’s ability to access the vaccine. Multivariate regression might be used in the future to control for these or other confounding variables. Further study may help determine whether the vaccine invitation strategy should differ based on demographic, geographic, or other factors (e.g., vaccine hesitancy, when known).

### 4.3. Future Directions

Due to the relatively low cost and low effort required, it is likely that invitation and reminder letters for unvaccinated women will be incorporated into routine operations in Manitoba. However, more research is needed to determine the optimal timing of the letters (e.g., age at which they should be mailed), how many letters/reminders to send, and the content of the invitation packages.

Additionally, more research is needed to investigate the impact of letters among different geographic regions in the province. For example, vaccination uptake was observed to be highest in the most urban health region and among individuals living in urban postal codes. This might be due to improved vaccine access in urban areas or different cultural beliefs, values, or preferences among people in those areas. Tailored interventions might be needed to improve HPV vaccine uptake in different areas of the province.

Further, alternate communication strategies (e.g., text or email messages) might be preferred by the targeted age group compared to mailed letters. Unfortunately, the screening registry does not contain personal phone numbers or email addresses, making direct digital communication impossible at this time. Digital awareness campaigns, such as social media or online advertising, could complement direct communication but could not be targeted towards unvaccinated individuals, as was the case in this study.

Finally, there were two anecdotal reports of people being unable to access HPV vaccines from their healthcare providers. This was resolved by providing one-on-one education to the providers and by redirecting the individuals to other provider options. Additional efforts may be needed to ensure knowledge among healthcare providers about the vaccine program, and, in particular, the once eligible, always eligible policy. Some of that education work is underway through new educational resources and continuing education programs.

## 5. Conclusions

Sending invitation and reminder letters to unvaccinated women may be a low-cost way to increase HPV vaccination coverage among the population eligible for the publicly funded immunization program. More research is needed to refine the invitation packages and determine the optimal timing and frequency of letters and whether the invitation strategy should differ based on demographic, geographic, or other factors.

## Figures and Tables

**Figure 1 curroncol-32-00216-f001:**
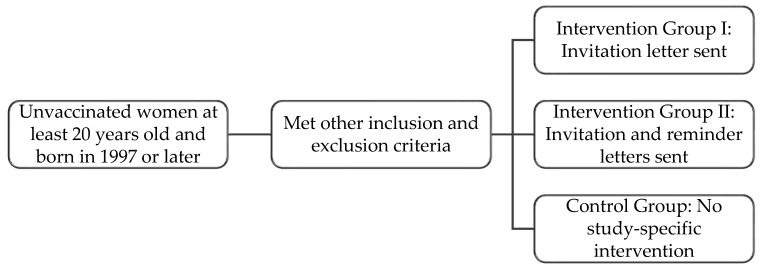
Process for identifying eligible individuals and separating them into three study groups.

**Figure 2 curroncol-32-00216-f002:**
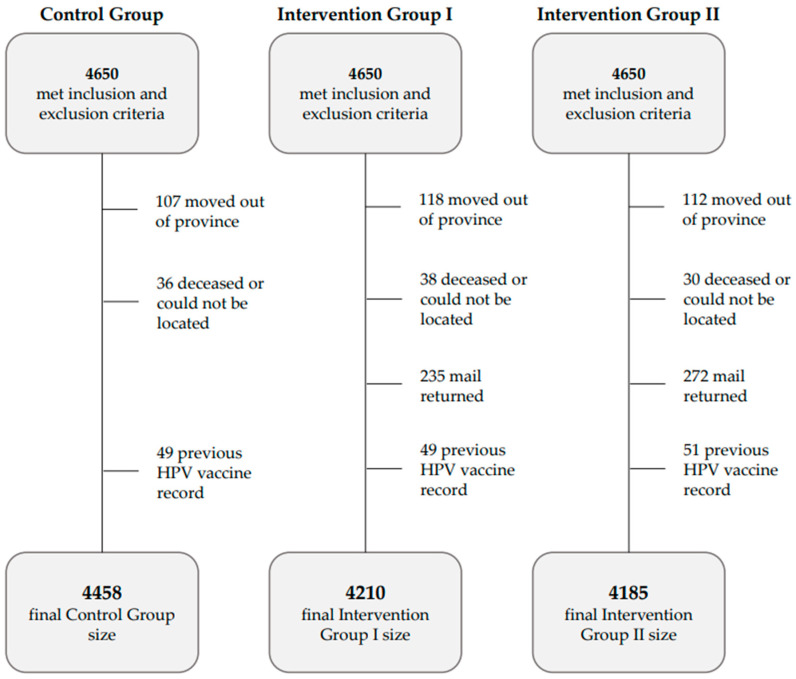
Final study group sizes and reasons for excluding individuals from analysis after randomization. The human papillomavirus vaccine is represented as HPV vaccine in table.

**Table 1 curroncol-32-00216-t001:** Cohort demographics.

	Control Group (No Study-Specific Letters)	Intervention Group I: Invitation Letter	Intervention Group II: Invitation Letter and Reminder Letter
Average age Age range	23.1 years 20–26 years	23.1 years 20–26 years	23.2 years 20–26 years
Living in a rural region of the province	23.4%	24.4%	25.5%
Unscreened for cervical cancer *	72.8%	74.3%	72.7%

* Pap testing is delivered through organized screening to women and eligible individuals aged 21–69. Unscreened is defined as having no record of ever having had a Pap test.

**Table 2 curroncol-32-00216-t002:** Human Papillomavirus (HPV) vaccine uptake among the control and intervention groups within six months of the invitation.

	Control Group (No Study-Specific Letters)	Intervention Group I: Invitation Letter	Intervention Group II: Invitation Letter and Reminder Letter
Group size (met inclusion and exclusion criteria over six months of follow-up)	4458	4210	4185
Received one dose of the HPV vaccine within six months of invitation sent date	38 (0.9%)	107 (2.5%)	168 (4.0%)
95% confidence interval	0.6–1.1%	2.1–3.0%	3.4–4.6%
*p*-value	<0.0001	<0.0001	<0.0001

**Table 3 curroncol-32-00216-t003:** Odds ratios with 95% confidence limits for six-month Human Papillomavirus (HPV) vaccine uptake.

	Control Group vs. Intervention group I: Invitation Letter	Control Group vs. Intervention Group II: Invitation and Reminder Letters
Odds ratio estimates	3.0	4.9
95% Wald confidence limits	2.1–4.4	3.4–6.9
*p*-value	<0.0001	<0.0001

**Table 4 curroncol-32-00216-t004:** Human papillomavirus (HPV) vaccine uptake among the control and intervention groups within 12 months of invitation.

	Control Group (No Study-Specific Letters)	Intervention Group I: Invitation Letter	Intervention Group II: Invitation Letter and Reminder Letter
Group size (met inclusion and exclusion criteria) over 12 months of follow-up	4430	4191	4167
Received one dose of the HPV vaccine within twelve months of invitation sent date	74 (1.7%)	157 (3.7%)	215 (5.2%)
95% confidence interval	1.3–2.1%	3.2–4.3%	4.5–5.8%
*p*-value	<0.0001	<0.0001	<0.0001

**Table 5 curroncol-32-00216-t005:** Odds ratios with 95% confidence limits for twelve-month Human Papillomavirus (HPV) vaccine uptake.

	Control Group vs. Intervention Group I: Invitation Letter	Control Group vs. Intervention Group II: Invitation and Reminder Letters
Odds ratio estimates	2.3	3.2
95% Wald confidence limits	1.7–3.0	2.5–4.2
*p*-value	<0.0001	<0.0001

**Table 6 curroncol-32-00216-t006:** Human Papillomavirus (HPV) vaccine uptake in urban and rural areas within 6 and 12 months of invitation.

Within 6 Months of Invitation	Control Group (No Study-Specific Letters)	Intervention Group I: Invitation Letter	Intervention Group II: Invitation Letter and Reminder Letter
**Urban residents**	3417	3183	3117
Received one dose of the HPV vaccine within 6 months of invitation	33 (1.0%)	89 (2.8%)	147 (4.7%)
**Rural residents**	1041	1027	1068
Received one dose of the HPV vaccine within 6 months of invitation	5 (0.5%)	18 (1.8%)	21 (2.0%)
*p*-value	0.1358	0.0647	<0.0001
**Within 12 months of invitation**	**Control Group (No Study-Specific Letters)**	**Intervention Group I:** **Invitation Letter**	**Intervention Group II:** **Invitation Letter and** **Reminder Letter**
**Urban residents**	3392	3168	3104
Received one dose of the HPV vaccine within 12 months of invitation	63 (1.9%)	127 (4.0%)	178 (5.7%)
**Rural residents**	1038	1023	1063
Received one dose of the HPV vaccine within 12 months of invitation	11 (1.1%)	30 (2.9%)	37 (3.5%)
*p*-value	0.0794	0.1150	0.0041

## Data Availability

The data associated with this study are not publicly available. This ensures the privacy and confidentiality of personal health information in accordance with Manitoba’s Personal Health Information Act. The data may be made available to health researchers following guidelines established by CancerCare Manitoba’s Research and Resource Impact Committee (https://www.cancercare.mb.ca/Research/research-office/research-impact-commitee accessed on 2 April 2025) and/or Manitoba Health’s Provincial Health Research Privacy Committee (https://www.rithim.ca/phrpc-overview accessed on 2 April 2025).

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
