# Peer review of "Using Invitation Letters to Increase HPV Vaccination Among Adult Women"

_curroncol, 2025, doi:10.3390/curroncol32040216_

Round 1
Reviewer 1 Report
Comments and Suggestions for Authors
This study addresses an important public health issue by exploring the effectiveness of mailed invitations and reminder letters in increasing HPV vaccination uptake among adult women in Manitoba. The manuscript is well-structured, with a clear introduction, methods, results, and discussion. However, some areas require clarification, further debate, or improvement.
- Abstract Clarity:
- The abstract should briefly clarify the findings' statistical significance and practical implications. Include confidence intervals and p-values in the results summary to strengthen the reported effect sizes.
- Another Example: “Significant gaps exist in their progress.” This is unclear phrasing. Suggested Improvement: Rephrase for clarity, e.g., “Significant gaps remain in HPV vaccination coverage among eligible adults.”
- Figures:
- Ensure all figures are clearly labeled and placed at the bottom of the figure.
- Grammar and Style:
- Minor grammatical inconsistencies should be addressed through proofreading. Example: "Individuals who received at least one dose of an approved HPV vaccine within six months of the invitation were counted as vaccinated." Consider specifying "were counted as vaccine adopters for analysis" for clarity.
- Justification for Outcome Measurement Period
- The study assesses vaccination uptake at six months and extends the analysis to 12 months. A stronger rationale for selecting these timeframes would be beneficial.
- Consider discussing whether a longer follow-up period might yield different or more comprehensive results.
- Consideration of Potential Confounders and External Factors
- The manuscript acknowledges that factors such as healthcare access, rurality, and healthcare-seeking behaviors were not controlled. Expanding on how these factors may have influenced the outcomes in the limitations section would improve the discussion.
- A stratified analysis based on geographic location (urban vs. rural) or healthcare access could provide deeper insights.
- Future studies could incorporate multivariable regression models to better control for confounding variables.
- Impact of Invitation Content and Delivery Methods
- While the study describes the involvement of public advisors in designing invitation letters, it does not evaluate the effectiveness of specific content elements. Were there variations in participant responses based on differences in messaging, wording, or format?
- Discussing the potential role of digital communication (e.g., emails, text messages) as a complement or alternative to mailed letters could enhance the analysis.
- Comparisons with Other Vaccination Campaigns
- The study would benefit from broader comparisons with successful HPV or other vaccine promotion strategies.
- Are there similar interventions in other regions or countries with varying effectiveness? A comparative analysis could provide valuable insights.
- Generalizability of Findings Beyond Manitoba
- Manitoba's unique “once eligible, always eligible” policy may limit the findings' applicability to other provinces or countries with stricter eligibility criteria.
- A more detailed discussion of how this policy affects generalizability would strengthen the manuscript.
- Policy Implications of the Findings
- The study highlights mailed invitation/reminder letters as a cost-effective intervention but does not fully explore how these findings can be implemented at a policy level.
- Consider discussing how this approach could be scaled up in other provinces or integrated into broader immunization programs.
- Potential Data Limitations
- The exclusion of women without valid Manitoba health insurance may underrepresent marginalized populations, such as recent immigrants or uninsured individuals.
- Acknowledging this limitation and suggesting strategies for future research to include these populations would enhance the study's robustness. (If possible).
- Discussion of Vaccine Accessibility Issues
- The study does not explore potential challenges in accessing HPV vaccines, despite anecdotal reports of difficulties in obtaining them.
- Including qualitative insights from healthcare providers or vaccine recipients could help identify barriers.
- Proposing system-level interventions to improve vaccine accessibility would be a valuable addition.
Conclusion
This manuscript presents an important study with significant public health implications for increasing HPV vaccine uptake. Strengthening the discussion on limitations, external validity, and alternative communication strategies would further enhance its impact.
Author Response
This study addresses an important public health issue by exploring the effectiveness of mailed invitations and reminder letters in increasing HPV vaccination uptake among adult women in Manitoba. The manuscript is well-structured, with a clear introduction, methods, results, and discussion. However, some areas require clarification, further debate, or improvement.
Thank you for taking the time to review the paper and for providing detailed and thorough feedback. We considered your comments carefully and have responded below (in blue) and in the revised manuscript through tracked changes.
- Abstract Clarity:
The abstract should briefly clarify the findings' statistical significance and practical implications. Include confidence intervals and p-values in the results summary to strengthen the reported effect sizes. Another Example: “Significant gaps exist in their progress.” This is unclear phrasing. Suggested Improvement: Rephrase for clarity, e.g., “Significant gaps remain in HPV vaccination coverage among eligible adults.”
- Thank you for noting this. P-values and confidence intervals have been added to the abstract.
- Several minor changes have been made to clarify the language in the paper. However, I am unable to find the referenced sentence ("Significant gaps remain ...") in the abstract (or elsewhere in the article). Please clarify where you saw that sentence.
- Figures:
Ensure all figures are clearly labeled and placed at the bottom of the figure.
- Figures 1 and 2 have been relabelled to more clearly describe the figures.
- We confirm that the figures are above their labels.
- Grammar and Style:
Minor grammatical inconsistencies should be addressed through proofreading. Example: "Individuals who received at least one dose of an approved HPV vaccine within six months of the invitation were counted as vaccinated." Consider specifying "were counted as vaccine adopters for analysis" for clarity.
- This change was made. Several additional edits were made throughout the paper to clarify language.
- Justification for Outcome Measurement Period
The study assesses vaccination uptake at six months and extends the analysis to 12 months. A stronger rationale for selecting these timeframes would be beneficial. Consider discussing whether a longer follow-up period might yield different or more comprehensive results.
- The selected timeframe was not based on published evidence but on typical local patterns. The rationale was clarified by adding additional language to section 2.3.
- We tested several timeframes up to one year and the uptake pattern persisted. We did not extend the timeframe beyond 12 months because we wanted to be able to attribute the behaviour (the vaccine appointment) to the intervention (receiving the letter). With longer timeframes, we became less confident that new confounders wouldn’t become relevant. We have added language to clarify the timeframes.
- Consideration of Potential Confounders and External Factors
The manuscript acknowledges that factors such as healthcare access, rurality, and healthcare-seeking behaviors were not controlled. Expanding on how these factors may have influenced the outcomes in the limitations section would improve the discussion. A stratified analysis based on geographic location (urban vs. rural) or healthcare access could provide deeper insights. Future studies could incorporate multivariable regression models to better control for confounding variables.
- Edits were made in section 4.2 to further describe the impact of potential confounders and how we might investigate them further in the future.
- A new section has been added to 3.4 to describe the differences in HPV vaccination uptake between urban and rural residents. This provides background for the comments in the discussion section.
- We agree that healthcare access (i.e., current use of healthcare services or proximity to a vaccine-delivery location) would provide deeper insights. Unfortunately, we do not have access to those data but will identify it as a possible future direction.
- We agree that future studies could include multivariate regression. Additional information has been added in 4.2 to clarify that as a future opportunity.
- Impact of Invitation Content and Delivery Methods
While the study describes the involvement of public advisors in designing invitation letters, it does not evaluate the effectiveness of specific content elements. Were there variations in participant responses based on differences in messaging, wording, or format? Discussing the potential role of digital communication (e.g., emails, text messages) as a complement or alternative to mailed letters could enhance the analysis.
- Everyone received the same invitation letter; there was no variation in message, design, or format during the study. We changed the order of that paragraph to clarify that the engagement process occurred first and clarified that the sent package was sent to everyone.
- We agree that digital communication is a significant potential opportunity. Unfortunately, digital communications is not an option within our current system, since our cervix registry does not contain complete contact information (phone numbers, email addresses). We have added a note to 4.3 to identify that digital communication is a consideration for the future.
- Comparisons with Other Vaccination Campaigns
The study would benefit from broader comparisons with successful HPV or other vaccine promotion strategies. Are there similar interventions in other regions or countries with varying effectiveness? A comparative analysis could provide valuable insights.
- We are not aware of studies that target young adults specifically. Most related research focuses on children (or their families), older adults, or specific sub-populations. We added a note about education and awareness activities campaigns in section 4.3.
- Generalizability of Findings Beyond Manitoba
Manitoba's unique “once eligible, always eligible” policy may limit the findings' applicability to other provinces or countries with stricter eligibility criteria. A more detailed discussion of how this policy affects generalizability would strengthen the manuscript.
- We have added a comment in section 4 that speaks to the generalizability of the findings. Many international jurisdictions provide free vaccination into young adulthood, which is the age group we considered.
- Policy Implications of the Findings
The study highlights mailed invitation/reminder letters as a cost-effective intervention but does not fully explore how these findings can be implemented at a policy level. Consider discussing how this approach could be scaled up in other provinces or integrated into broader immunization programs.
- As above, we added a comment in section 4 about how these results could be used in other jurisdictions. We are currently considering how this work could be incorporated into broader immunization programs, but do not have any update at this time. For now, we have referenced this as an area for future study.
- Potential Data Limitations
The exclusion of women without valid Manitoba health insurance may underrepresent marginalized populations, such as recent immigrants or uninsured individuals. Acknowledging this limitation and suggesting strategies for future research to include these populations would enhance the study's robustness. (If possible).
- A comment has been added to the introduction to clarify who is eligible for vaccine coverage in the province. Coverage is broad and covers the entire population living in the province, including recent immigrants.
- Discussion of Vaccine Accessibility Issues
The study does not explore potential challenges in accessing HPV vaccines, despite anecdotal reports of difficulties in obtaining them. Including qualitative insights from healthcare providers or vaccine recipients could help identify barriers. Proposing system-level interventions to improve vaccine accessibility would be a valuable addition.
- We have clarified that there were only two anecdotal reports and described how they were addressed. Work is underway to address system barriers but it has not yet been evaluated.
Conclusion
This manuscript presents an important study with significant public health implications for increasing HPV vaccine uptake. Strengthening the discussion on limitations, external validity, and alternative communication strategies would further enhance its impact.
Reviewer 2 Report
Comments and Suggestions for Authors
1-Good presentation, however, in the abstract, the control group was not defined.
2-The materials and methods, the control group requires to be detailed, i.e. how many individuals received the complete three doses, two doses, and one dose of HPV vaccine? While in the results section, As in the line 137-138, described: Of the individuals who only received the invitation letter, 2.5% received had at least one dose of the HPV vaccine, compared to only 0.9% of the control group, the 9% was not defined in materials and methods.
The results and discussion are well-described.
Comments on the Quality of English Language
Good presentation, however, in the abstract, the control group was not defined.
The materials and methods, the control group requires to be detailed, i.e. how many individuals received the complete three doses, two doses, and one dose of HPV vaccine? While in the results section, As in the line 137-138, described: Of the individuals who only received the invitation letter, 2.5% received had at least one dose of the HPV vaccine, compared to only 0.9% of the control group, the 9% was not defined in materials and methods.
The results and discussion are well-described.
Author Response
Thank you for taking the time to review the paper and for providing your feedback. We considered your comments carefully and have responded below (in blue) and in the revised manuscript through tracked changes.
1-Good presentation, however, in the abstract, the control group was not defined.
- Thank you for noting that. A reference to the control group was added to the abstract.
2-The materials and methods, the control group requires to be detailed, i.e. how many individuals received the complete three doses, two doses, and one dose of HPV vaccine? While in the results section, As in the line 137-138, described: Of the individuals who only received the invitation letter, 2.5% received had at least one dose of the HPV vaccine, compared to only 0.9% of the control group, the 9% was not defined in materials and methods.
- Provincial guidelines recommend that doses be given at least 6 months apart for that age group, so we would not expect more than one dose in the 6-month follow-up timeframe. We have edited the language in 3.2 and 4 to “one dose” to provide clarity.
- The control group has been defined in section 2.1
The results and discussion are well-described.
Reviewer 3 Report
Comments and Suggestions for Authors
The study investigates whether sending invitation and reminder letters can improve HPV vaccine uptake among eligible but unvaccinated adult women in Manitoba, Canada. The study randomized 14,000 women into three groups: two intervention groups receiving invitation letters (with one group also receiving reminder letters) and a control group receiving no correspondence. The results showed that the intervention groups had significantly higher vaccination rates compared to the control group, with the invitation and reminder group having the highest uptake. The study concludes that such low-cost interventions could be effective in increasing HPV vaccination coverage among eligible adults.
While the study addresses an important public health issue related to HPV vaccination, there are several critical concerns that undermine its validity and reliability.
#1 Low Overall Vaccination Uptake: Despite the significant differences between the intervention and control groups, the overall vaccination uptake remains very low (4.0% in the invitation/reminder group and 2.5% in the invitation-only group). This suggests that the intervention may not be practically effective on a larger scale, as it only marginally increases vaccination rates. #2 Lack of Control for Confounding Variables: The study did not control for potential confounding factors such as socioeconomic status, education level, access to healthcare services, or prior vaccination attitudes. These factors are known to influence vaccination uptake and could have confounded the observed results. Without controlling for these variables, it is difficult to attribute the observed increase in vaccination solely to the invitation and reminder letters. #3 Sample Selection and Data Quality Issues: The study relied on data from the CervixCheck registry, which had issues with delayed data entry and incorrect addresses, leading to the exclusion of participants after randomization. This raises concerns about the accuracy and completeness of the data used in the analysis. Additionally, the high number of returned mail packages (over 500) due to incorrect addresses suggests potential issues with the representativeness of the sample. #4 Limited Generalizability: The study was conducted in a specific geographic region (Manitoba) with a unique policy of "once-eligible, always-eligible" for HPV vaccination. This limits the generalizability of the findings to other regions with different policies or healthcare systems. The results may not be applicable to areas with higher population density or different healthcare access challenges. #5 Short Follow-Up Period: Although the study extended the follow-up period to 12 months, the initial six-month period may still be too short to capture the full impact of the intervention. HPV vaccination decisions might be influenced by longer-term factors, and a longer follow-up period could provide more robust results. #6 Lack of Qualitative Insights: The study is purely quantitative and lacks qualitative data on why the invitation and reminder letters were effective or not. Understanding the participants' perceptions, barriers, and motivations could provide valuable insights and improve the design of future interventions. While the study's findings suggest that invitation and reminder letters may have some impact on HPV vaccination uptake, the overall effectiveness is limited, and the methodological limitations significantly weaken the study's robustness and generalizability. Given these concerns, the study does not provide sufficient evidence to support the conclusion that such interventions are a viable solution for increasing HPV vaccination coverage. Therefore, it is recommended that this draft be substantial revised or rejected if those flaws cannot be thoroughly addressed.Author Response
The study investigates whether sending invitation and reminder letters can improve HPV vaccine uptake among eligible but unvaccinated adult women in Manitoba, Canada. The study randomized 14,000 women into three groups: two intervention groups receiving invitation letters (with one group also receiving reminder letters) and a control group receiving no correspondence. The results showed that the intervention groups had significantly higher vaccination rates compared to the control group, with the invitation and reminder group having the highest uptake. The study concludes that such low-cost interventions could be effective in increasing HPV vaccination coverage among eligible adults.
While the study addresses an important public health issue related to HPV vaccination, there are several critical concerns that undermine its validity and reliability.
Thank you for taking the time to review the paper and for providing detailed feedback. We considered your comments carefully and have responded below (in blue) and in the revised manuscript through tracked changes.
#1 Low Overall Vaccination Uptake: Despite the significant differences between the intervention and control groups, the overall vaccination uptake remains very low (4.0% in the invitation/reminder group and 2.5% in the invitation-only group). This suggests that the intervention may not be practically effective on a larger scale, as it only marginally increases vaccination rates.
- It is true that the overall uptake was low, but we do not believe that the low uptake undermines the validity or reliability of the results. It may, however, impact the practical utility of the findings. We have commented on this in sections 4.2. and 4.3.
#2 Lack of Control for Confounding Variables: The study did not control for potential confounding factors such as socioeconomic status, education level, access to healthcare services, or prior vaccination attitudes. These factors are known to influence vaccination uptake and could have confounded the observed results. Without controlling for these variables, it is difficult to attribute the observed increase in vaccination solely to the invitation and reminder letters.
- We agree that this study would have benefited from controlling for potential confounding variables. This limitation was addressed in section 4.2 and has been expanded upon. We note that future studies could include multivariate regression. Additional information has been added in 4.2 to clarify that as a future opportunity.
- We agree that an analysis of other factors would provide deeper insights. Unfortunately, we do not have access to those data but have identified it as a possible future direction.
- A new section has been added to 3.4 to describe the differences in HPV vaccination uptake between urban and rural residents. This provides background for the comments in the discussion section.
#3 Sample Selection and Data Quality Issues: The study relied on data from the CervixCheck registry, which had issues with delayed data entry and incorrect addresses, leading to the exclusion of participants after randomization. This raises concerns about the accuracy and completeness of the data used in the analysis. Additionally, the high number of returned mail packages (over 500) due to incorrect addresses suggests potential issues with the representativeness of the sample.
- Edits were made to section 2.2 to help clarify the exclusions
- About 50 people in each group were removed because of delayed data entry. Unfortunately, this is a reality of public health data, as services are provided every day in the health system but are not submitted “live” to the database (providers have six months to submit data).
- Many people were removed during the 6-month follow-up because they died or moved out of the province. Unfortunately, it is not possible to design a study that will avoid this.
- Returned mail is a reality of our health system data. We have added a comment in section 4.2 to hypothesize the reason for the high return mail rate.
#4 Limited Generalizability: The study was conducted in a specific geographic region (Manitoba) with a unique policy of "once-eligible, always-eligible" for HPV vaccination. This limits the generalizability of the findings to other regions with different policies or healthcare systems. The results may not be applicable to areas with higher population density or different healthcare access challenges.
- We agree that Manitoba is somewhat unique, but the study may be relevant to any jurisdiction that vaccinates into young adulthood. A comment regarding the generalizability of the results was added in section 4.
#5 Short Follow-Up Period: Although the study extended the follow-up period to 12 months, the initial six-month period may still be too short to capture the full impact of the intervention. HPV vaccination decisions might be influenced by longer-term factors, and a longer follow-up period could provide more robust results.
- Although extending the follow-up period may yield higher vaccination rates, it would likely be difficult to attribute the behaviour (the vaccine appointment) to the intervention (receiving the letter). With longer timeframes, we became less confident that new confounders wouldn’t become relevant. We have added language to clarify the timeframes.
- The selected timeframe was not based on published evidence but on typical local patterns. The rationale was clarified by adding additional language to section 2.3.
- Vaccines in Manitoba are available quickly and therefore would not require more than 12 months of follow-up to observe associated behaviour.
#6 Lack of Qualitative Insights: The study is purely quantitative and lacks qualitative data on why the invitation and reminder letters were effective or not. Understanding the participants' perceptions, barriers, and motivations could provide valuable insights and improve the design of future interventions.
- We agree that a qualitative analysis would have provided additional insight. We have added comments in 4.2 and 4.3 regarding the potential benefits of understanding participants’ perceptions, barriers and motivations.
Reviewer 4 Report
Comments and Suggestions for Authors
HPV vaccination (especially when administered to pre-adolescent subjects) is safe and highly effective in preventing HPV infection and related clinical manifestations, including carcinomas. High coverage ensures the maximum benefit to the vaccinees and the population at large. Efforts to increase coverage are being undertaken in several countries, and are instrumental to reach the WHO goal of cervical cancer elimination. In this manuscript, the authors report the results of a study conducted in Manitoba (Canada), where a publicly-funded school-based HPV vaccine program has being offered to girls born in 1997 or later when they attend school grade six. Since in this region a once-eligible, always-ligible policy is in place, they evaluated the impact of sending an invitation letter (group 1) or invitation+reminder letters (group 2) to women at least 20 yrs-old who did not adhere to the program, in comparison to a control group. Apprximately 14000 women were involved in the study; over the following six months, at least one vaccine dose was administered in 2.5% and 4.0% women enrolled in intervention groups 1 and 2, respectively, in contrast to 0.9% of women in the control group. The authors conclude that the evaluated intervention is feasible and can increase the vaccination coverage. The data well support the conclusions, and the paper is well written (there are just a few typing errors to be looked at). Some suggestions to further improve the manuscrips are reported below.
-MATERIALS AND METHODS: some information on the cervical screening policy in place (organized or opportunistic screening? starting age?) could be added.
-RESULTS, Tables 2 and 4: also 0.9% and 1.7% of women of the control group received at least one dose of HPV vaccine within six or twelve months of the intervention, respectively. Are these figures in line with what observed without any intervention? Or could this intervention have influenced also these women, i.e. because an information campaign was conducted, or because the information was spread by the women of the intervention group?
-DISCUSSION, 4.3 Future directions: could it be forecasted to implement this intervention on eligible women younger than 20 yrs? Vaccination of girls younger than 17 has been demonstrated to be more effective than vaccination beyond that age.
Author Response
HPV vaccination (especially when administered to pre-adolescent subjects) is safe and highly effective in preventing HPV infection and related clinical manifestations, including carcinomas. High coverage ensures the maximum benefit to the vaccinees and the population at large. Efforts to increase coverage are being undertaken in several countries, and are instrumental to reach the WHO goal of cervical cancer elimination. In this manuscript, the authors report the results of a study conducted in Manitoba (Canada), where a publicly-funded school-based HPV vaccine program has being offered to girls born in 1997 or later when they attend school grade six. Since in this region a once-eligible, always-ligible policy is in place, they evaluated the impact of sending an invitation letter (group 1) or invitation+reminder letters (group 2) to women at least 20 yrs-old who did not adhere to the program, in comparison to a control group. Apprximately 14000 women were involved in the study; over the following six months, at least one vaccine dose was administered in 2.5% and 4.0% women enrolled in intervention groups 1 and 2, respectively, in contrast to 0.9% of women in the control group. The authors conclude that the evaluated intervention is feasible and can increase the vaccination coverage. The data well support the conclusions, and the paper is well written (there are just a few typing errors to be looked at). Some suggestions to further improve the manuscrips are reported below.
Thank you for taking the time to review the paper and for providing feedback. We considered your comments carefully and have responded below (in blue) and in the revised manuscript through tracked changes.
-MATERIALS AND METHODS: some information on the cervical screening policy in place (organized or opportunistic screening? starting age?) could be added.
- Thank you for noting this. Information about the cervical screening program has been added to section 2.1 and Table 1.
-RESULTS, Tables 2 and 4: also 0.9% and 1.7% of women of the control group received at least one dose of HPV vaccine within six or twelve months of the intervention, respectively. Are these figures in line with what observed without any intervention? Or could this intervention have influenced also these women, i.e. because an information campaign was conducted, or because the information was spread by the women of the intervention group?
- We did not run a campaign during this time, but regular education events occurred. We have added a comment in 4.1 related to this.
-DISCUSSION, 4.3 Future directions: could it be forecasted to implement this intervention on eligible women younger than 20 yrs? Vaccination of girls younger than 17 has been demonstrated to be more effective than vaccination beyond that age.
- We agree that vaccination at younger ages is most beneficial. However, there is increasing evidence that the HPV vaccine can be effective when given to adults. We wanted to study the impact of reaching people (adults) who could make their own healthcare decisions (vs. their parents or guardians).
- Unfortunately, the screening registry does not contain information about children (or their families). A comment has been added to section 4.2 to describe this limitation
Round 2
Reviewer 3 Report
Comments and Suggestions for Authors
Most of the issues have been addressed but "the high number of returned mail" really undermined the validity or reliability of the results. This should be avoid with well-performed preparedness. The authors should acknowledge the limitation and remind the readers to uptake their conclusion with cautions.
Author Response
Most of the issues have been addressed but "the high number of returned mail" really undermined the validity or reliability of the results. This should be avoid with well-performed preparedness. The authors should acknowledge the limitation and remind the readers to uptake their conclusion with cautions.
Returned mail cannot be avoided within our registry, as there is no requirement for the public to routinely register their contact information with the insurer. Unless/until they seek care at certain public facilities, insured individuals continue to receive health care services, even if their address is recorded incorrectly. The information is coming directly from the provincial source-of-truth, there is no other demographic database to pull from.
To address your concerns, we have further described “returned mail” in section 2.2 and provided an example of a return rate in a similar population in a similar timeframe. An additional comment has also been added to 4.2 to further describe the limitation.